# Field-free spin-orbit torque-induced switching of perpendicular magnetization in a ferrimagnetic layer with a vertical composition gradient

Zhenyi Zheng[1,2,3,7], Yue Zhang[1,7✉], Victor Lopez-Dominguez[2], Luis Sánchez-Tejerina[4], Jiacheng Shi[2], Xueqiang Feng[1], Lei Chen[1], Zilu Wang[1], Zhizhong Zhang[1], Kun Zhang[1], Bin Hong[1], Yong Xu[1], Youguang Zhang[3], Mario Carpentieri[5], Albert Fert[1,6], Giovanni Finocchio[4✉], Weisheng Zhao[1✉] & Pedram Khalili Amiri[2✉]

Current-induced spin-orbit torques (SOTs) are of interest for fast and energy-efficient manipulation of magnetic order in spintronic devices. To be deterministic, however, switching of perpendicularly magnetized materials by SOT requires a mechanism for in-plane symmetry breaking. Existing methods to do so involve the application of an in-plane bias magnetic field, or incorporation of in-plane structural asymmetry in the device, both of which can be difficult to implement in practical applications. Here, we report bias-field-free SOT switching in a single perpendicular CoTb layer with an engineered vertical composition gradient. The vertical structural inversion asymmetry induces strong intrinsic SOTs and a gradient-driven Dzyaloshinskii–Moriya interaction (g-DMI), which breaks the in-plane symmetry during the switching process. Micromagnetic simulations are in agreement with experimental results, and elucidate the role of g-DMI in the deterministic switching processes. This bias-field-free switching scheme for perpendicular ferrimagnets with g-DMI provides a strategy for efficient and compact SOT device design.

[1] Fert Beijing Research Institute, School of Integrated Circuit Science and Engineering, Beijing Advanced Innovation Center for Big Data and Brain Computing, Beihang University, Beijing, PR China. [2] Department of Electrical and Computer Engineering, Northwestern University, Evanston, IL, USA. [3] School of Electronics and Information Engineering, Beihang University, Beijing, PR China. [4] Department of Mathematical and Computer Sciences, Physical Sciences and Earth Sciences, University of Messina, Messina, Italy. [5] Dipartimento di Ingegneria Elettrica e dell'Informazione, Politecnico di Bari, Bari, Italy. [6] Unité Mixte de Physique, CNRS, Thales, Université Paris-Sud, Université Paris-Saclay, Palaiseau, France. [7]These authors contributed equally: Zhenyi Zheng, Yue Zhang. ✉email: yz@buaa.edu.cn; gfinocchio@unime.it; weisheng.zhao@buaa.edu.cn; pedram@northwestern.edu

Spin-orbit torque (SOT) is a leading contender as a fast and low-power method to manipulate magnetic order in spintronic devices, particularly for artificial intelligence and high-performance computing applications where high-speed on-chip memory is required[1–4]. SOT switching of ferrimagnetic (FIM) materials, in particular, is of great current interest[5–9]. FIMs exhibit the exchange-dominated high-frequency (sub-terahertz) dynamics of antiferromagnets, which can result in high switching speed[5–7], while avoiding the difficulties of controlling domain size commonly encountered in antiferromagnets[4]. They can also utilize the relatively straightforward electrical readout methods available in ferromagnetic material systems (due to their non-zero magnetization)[7–9]. Hence, FIMs are promising as near-term candidates for practical high-performance SOT devices.

Typically, for memory applications where information is encoded in the direction of the magnetization in magnets with perpendicular magnetic anisotropy (PMA), SOT switching is realized in a non-magnet/ferromagnet (NM/FM) or NM/FIM heterostructure with the assistance of an external in-plane magnetic field $H_{ex}$ along the current direction[1,2,10–12]. SOTs originating from the spin Hall effect (SHE) in the NM (e.g., heavy metals[2,10] or topological insulators[11,12]) or from the Rashba effect[1,13] at the NM/FM interface can drive the adjacent magnetization to switch, while $H_{ex}$ breaks the in-plane inversion symmetry and leads to a deterministic switching direction for a particular direction of the in-plane current.

The requirement of external $H_{ex}$ in switching, however, hinders the integration of SOT devices on semiconductor chips. Thus, mechanisms to break the in-plane (structural or magnetic) symmetry, instead of $H_{ex}$, are being investigated to realize practical SOT memory devices[13–28]. To date, these have included in-plane exchange bias fields[14–16], interlayer exchange coupling[17,18], in-plane structural, composition, or interfacial oxidation asymmetry[19–25], and combining multiple competing sources of spin torque in one device[26–28]. However, with the exception of methods combining more than one source of torque, these schemes all rely on breaking the static in-plane symmetry in the device. In many cases, this makes it difficult to build large arrays with uniform device properties on the same wafer.

Here we experimentally show, instead, that a structure featuring in-plane symmetry, along with inversion asymmetry only along the growth direction, can also exhibit deterministic switching due to the dynamic in-plane breaking of symmetry during the switching process, induced by the Dzyaloshinskii–Moriya interaction (DMI).

The DMI effect can induce spin textures (e.g., domain walls and skyrmions) with broken chiral symmetry[29–32]. Therefore, DMI has recently been theoretically proposed as a symmetry-breaking mechanism that, in combination with damping-like and field-like SOTs, may enable deterministic switching in the absence of an external field[33–35]. However, field-free deterministic switching due to the combined action of DMI and SOT has not been reported to date.

Our approach to engineer the required DMI is inspired by a recent work reporting a large bulk DMI in a rare-earth (RE)-transition-metal (TM) ferrimagnet (FIM), where a vertical composition gradient was brought about by the natural inhomogeneous elemental composition distribution in alloys with large thickness[36]. Here, in order to maximize the DMI, we instead engineer a vertical composition gradient in the FIM layer by depositing ultrathin individual layers with a varying RE/TM composition ratio, leading to a broken inversion symmetry along the growth direction. This results in a gradient-driven DMI (g-DMI) which has the same symmetry as interfacial DMI in previously studied NM/FIM and NM/FM systems, which, however, is present throughout the bulk of the FIM due to the continuous composition gradient.

In addition, as RE elements possess large spin-orbit coupling (SOC) due mainly to their 5d band[37,38], they can generate spin-currents as large as those generated in nonmagnetic 5d metals such as Pt at the opposite end of the 5d metal series[39]. The 3d electrons of cobalt possess a much smaller SOC and the f electrons of Tb do not participate in the conduction. The combination of large spin-orbit-coupling and broken inversion symmetry generates Rashba interactions[40], which give rise to current-induced Rashba spin polarizations and resulting spin currents[41] which, in turn, can produce a SOT on the FIM magnetization, as discussed in the next section.

Using the combination of SOT and g-DMI effects, we demonstrate efficient field-free SOT magnetization switching in a single ferrimagnetic CoTb layer with a vertical composition gradient. Experiments and simulations indicate that this scheme can eliminate the need for NM and $H_{ex}$ simultaneously in the previously mentioned SOT switching systems. The field-free switching data are in agreement with micromagnetic simulations. Our work provides a pathway towards field-free SOT switching of perpendicular ferrimagnets which can potentially be scaled to large wafer size with good uniformity.

## Results

**Large SOT originating from the Tb composition gradient**. As a ferrimagnetic alloy, in which RE elements and TM elements are coupled anti-ferromagnetically, CoTb can exhibit robust PMA in a wide composition range[8,9]. To investigate the exact composition range of PMA, a series of Al (2 nm)/$Co_{1-a}Tb_a$ (6 nm)/Al (3 nm) films (see Fig. 1a) were deposited on thermally oxidized silicon substrates, with $a$ being the concentration of Tb in the alloy. Here, Al was chosen due to its negligible SOT. CoTb was constructed by a co-sputtering process of Co and Tb at different powers to control the stoichiometry of the final CoTb layer. As shown in Fig. 1b, $Co_{0.87}Tb_{0.13}$ and $Co_{0.56}Tb_{0.44}$ were the most Co-rich and the most Tb-rich samples which provided good PMA properties in this heterostructure. In addition, by the fitting of the net saturation magnetization $M_s$ dependence on composition, $Co_{0.71}Tb_{0.29}$ was determined as the magnetization compensation point at room temperature.

Guided by this information, samples with a vertical composition gradient were deposited next. As illustrated in Fig. 1c, the ferrimagnet consists of six layers of CoTb, each having a thickness of 1 nm. Here, we fixed the bottom layer composition as $Co_{0.87}Tb_{0.13}$, while the other five deposited layers followed a certain Tb composition step $\delta$, resulting in a nominal gradient of $\delta$ per nanometer. Hence, their composition can be expressed as $Co_{0.87-n\delta}Tb_{0.13+n\delta}$, where $n = 1, 2, 3, 4, 5$. Figure 1d shows that increasing the composition gradient leads to a decrease of the overall $M_s$ of the sample, which corresponds to the Co-rich part tendency in Fig. 1b, i.e., increasing the average Tb composition makes the sample approach the magnetic compensation point[8,9]. When $\delta$ arrives at 0.07, the sample is nearly compensated ($M_s < 50$ emu/cc), while the sample starts to show an in-plane magnetization component when exceeding $\delta = 0.07$. We thus only discuss gradients $\delta$ in the range from 0 to 0.07. To verify the existence of and quantify the composition gradient in our samples, high-resolution cross-sectional scanning transmission electron microscopy (STEM) and energy-dispersive X-ray spectroscopy (EDS) were performed on the sample with $\delta = 0.07$. Figure 1e shows the structure of the ordered layers in the sample. No obvious crystal structure is observed, revealing the amorphous nature of the CoTb layer as expected. EDS curves versus the thickness direction in Fig. 1f exhibit a prominent inhomogeneous spatial distribution of Co and Tb elements. With the measured position penetrating to the substrate, Co (Tb)

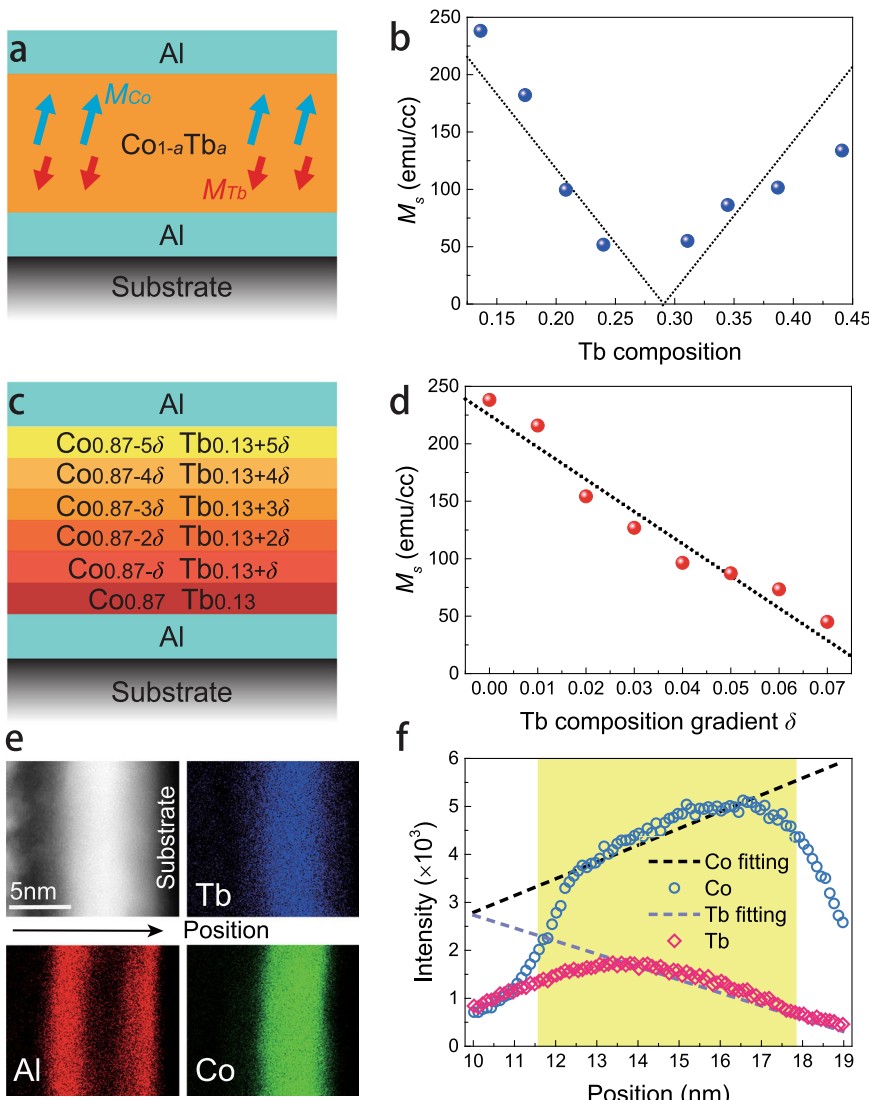

**Fig. 1 Structure and characterization of CoTb films with a vertical composition gradient. a** Structure of the homogenous $Co_{1-a}Tb_a$ samples. **b** Net magnetization dependence on the Tb concentration in homogenous CoTb samples. A clear magnetization compensation point is found at the point where Tb concentration equals 0.29. **c** Structure of the CoTb samples with composition gradient $\delta$. **d** Net magnetization dependence on $\delta$, showing a linear decrease. **e** STEM images of the sample with $\delta = 0.07$. **f** EDS-measured Co and Tb intensity in the sample as a function of vertical position. Co and Tb concentrations show opposite slopes, which verifies the existence as well as the direction of the composition gradient.

intensity shows an increasing (decreasing) tendency, which corresponds well with the designed structure in Fig. 1c.

To perform electrical measurements, the films were fabricated into Hall bar devices with a width of 10 μm by conventional lithography and ion milling. The inset in Fig. 2a displays the device as well as the measurement contact configurations. Based on the anomalous Hall effect resistance ($R_{AHE}$) measurements, all samples show clear PMA consistent with the film-level magnetometry results. The case of $\delta = 0.06$ is shown as an example in Fig. 2a. Note that with $\delta$ varying from 0.00 to 0.07, the $R_{AHE}$ loop does not change its polarity, revealing that all samples remain Co-rich overall.

Standard harmonic measurements were then carried out in a large field range to determine the SOT efficiency as well as the effective spin Hall angle of each sample[42,43]. When an alternating current $j = j_0 sin(\omega t)$ is applied along the $x$ axis, the SOT-induced alternating effective field will generate magnetization oscillations around the equilibrium position, which contributes to the rise of a second-harmonic Hall resistance $R_{2\omega}$. In the regime where the

in-plane magnetic field $|H_x|$ exceeds the magnetic anisotropy field $H_k$, $R_{2\omega}$ can be described by the equation[42,43]

$$R_{2\omega} = \frac{1}{2}\frac{R_{AHE}H_{DL}}{|H_x| - H_k} + R_{offset},$$ (1)

where $H_{DL}$ and $R_{offset}$ are the damping-like SOT effective field and resistance offset, respectively. Figure 2b, c plot the first harmonic resistance $R_\omega$ and $R_{2\omega}$ of the sample with $\delta = 0.06$. By fitting Fig. 2c with Eq. (1), the SOT efficiency $\xi = H_{DL}/j_{CoTb}$, where $j_{CoTb}$ is the corresponding current density in the CoTb layer, is determined to be $3.2 \pm 0.3$ Oe/($10^{10}$ A/m$^2$). The calculation of $j_{CoTb}$ can be found in Supplementary Note S1.

We then calculated the damping-like effective spin Hall angle by $\theta_{DL} = 2eM_s t_{CoTb} H_{DL}/\hbar j_{CoTb}$, where $t_{CoTb}$ is the CoTb layer thickness. The $\theta_{DL}$ values of samples with different gradients $\delta$ are summarized in Fig. 2d. Interestingly, $\theta_{DL}$ shows a clear increasing tendency as a function of $\delta$. At $\delta = 0.07$, the maximum value of

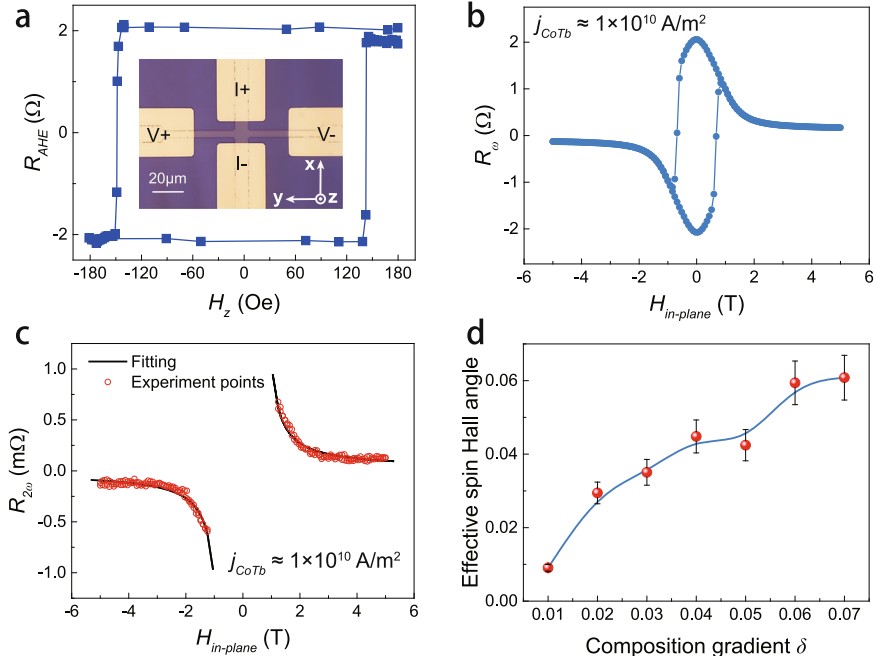

**Fig. 2 Electrical measurement setup and SOT characterization. a** Measured AHE loop for the sample with $\delta = 0.06$. The square shape reveals that the samples are perpendicularly magnetized. Inset shows a photograph of a representative Hall bar device. **b** $R_\omega$ versus magnetic field along the $x$ direction, for the sample with $\delta = 0.06$. **c** $R_{2\omega}$ and the fitting line versus magnetic field along the $x$ direction for the sample with $\delta = 0.06$ under $j_{CoTb} \approx 1 \times 10^{10}$ A/m$^2$. **d** Increasing tendency of $\theta_{DL}$ versus composition gradient $\delta$. Error bars are calculated from $R_{2\omega}$ measurements under different current densities for each sample.

$\theta_{DL}$ is determined to be $0.061 \pm 0.006$, which is comparable to the reported effective spin Hall angle in Pt[10].

The appearance of this sizeable SOT is a consequence of the combined large spin-orbit-coupling and broken inversion symmetry in our structure, which give rise to current-induced Rashba spin polarizations and resulting spin currents[40,41]. The breaking of the inversion symmetry in a magnetic layer can result from nonsymmetric interfaces, as in the classical model of Manchon and Zhang[44], or from composition gradients as in our samples, which can be called "g-Rashba" by analogy with "g-DMI" coined above for DMI. In a magnetic or ferrimagnetic material, the Rashba spin polarization and the resulting spin currents present a combination of terms of spin anomalous hall effect (SAHE) symmetry (spin-polarized along the axis of the magnetization, $M$) and SHE symmetry (spin-polarized along the $y$ axis for current along $x$ and emission along $z$), see Note S1 in[39] for Rashba-induced spin currents and the more general theory by Stiles and coworkers[45,46].

The SAHE term, polarized along with the magnetization $M$, cannot generate a torque on $M$. The SHE-like spin current in a magnetic layer can generate a Damping-Like (DL) spin transfer torque on $M$ if there is a possible transfer of the spins carried by this current, either from the magnetic layer to outside[41,47] or between different parts of the magnetic layer. Spin transfer from the magnetization of the CoTb layer to outside cannot work in our samples because the CoTb layer is inserted between two thin layers of the light element and weak absorber Al. In contrast, the asymmetric distribution of CoTb layers of different compositions, with higher 5d concentrations and larger Rashba spin polarization at the top, can give rise to unbalanced spin transfers between different parts of the sample. In our samples, there should be some imbalance between the transfer to the bottom magnetization of the larger downward spin currents $j_{CoTb}\theta_{DL,top}$ emitted from top and the transfer to the top of the smaller upward spin

currents $j_{CoTb}\theta_{DL,bottom}$ emitted with opposite spin polarization from the bottom. The final torque is difficult to predict precisely because it will depend on how the gradient of Tb concentration is reflected in the variation of both the Rashba interaction and other parameters, such as the spin dephasing length, controlling the proportion of spin transfer. In rough approximation, the DL torque should correspond to the transfer of a spin current of the order of $j_{CoTb}(\theta_{DL,top} - \theta_{DL,bottom})$ (the bracket corresponds to an effective spin Hall angle $\theta_{DL}$ which, as expected, is zero if the layer is homogeneous).

**Field-free switching of perpendicular CoTb by SOT and gradient-driven DMI.** We next performed SOT-induced magnetization switching experiments on the CoTb samples with a vertical composition gradient. Following the measurement strategy shown in Fig. 3a, successive write current pulses with a duration of 0.1 ms (orange bars, current from positive to negative and then back to positive) were injected in the $x$ axis of the Hall bar, while a small sensing current of 0.1 mA (blue triangles) was applied after each pulse to detect the magnetization state via $R_{AHE}$. Figure 3b shows $R_{AHE}$ as a function of the injected write current density $j_{CoTb}$ in the sample with $\delta = 0.07$, under a bias field $H_{ex}$ along the $x$ axis varying from 100 Oe to $-100$ Oe. The successful magnetization switching verifies the existence of intrinsic SOTs in the CoTb with composition gradient. Meanwhile, the current-driven $R_{AHE}$ shows a similar amplitude to the field-driven $R_{AHE}$, revealing that a nearly complete switching of the CoTb is achieved by the current. We define the critical switching current $j_c$ as the value of $j_{CoTb}$ where half of the maximum resistance change is achieved. From Fig. 3b, the $j_c$ of this 6 nm thick sample was determined to be around $9 \times 10^{10}$ A/m$^2$. An analysis of the $H_{ex}$ dependence of $j_c$ can be found in Supplementary Note S2.

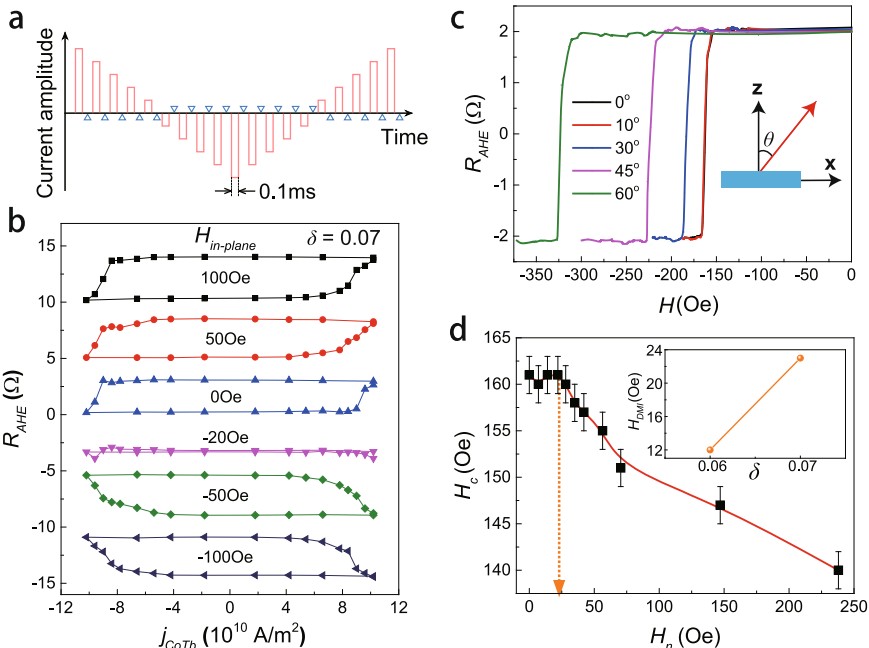

**Fig. 3 SOT switching experiments and DMI measurement. a** Measurement scheme of SOT switching. Writing pulses with a width of 0.1 ms were applied from positive to negative, and then back to positive. After each writing pulse, a small read current was applied to detect the magnetic state of the CoTb layer by AHE. **b** SOT switching curves of the sample with $\delta = 0.07$ under different in-plane magnetic fields (including zero-field). **c** AHE curves when the magnetization switches from up to down under different magnetic field angles $\theta$ with respect to the film normal direction. The inset shows the definition of $\theta$ in the $xz$ plane. **d** $H_c$ as a function of $H_n$. There is a threshold value of $H_n$ ($\approx 23$ Oe), above which $H_c$ starts to decrease. Error bars are obtained from repeated measurements for each $\theta$. The inset shows the increasing relationship between $H_{DMI}$ and $\delta$ in 6 nm CoTb films.

Supplementary Note S8 compares the SOT switching efficiency of these CoTb gradient samples—defined as $j_c/(t \cdot K_{eff})$, where $t$ is the magnetic film thickness and $K_{eff}$ is its effective perpendicular anisotropy energy density— with: (i) previous reports of SOT switching of ferrimagnetic CoTb films (in the presence of a bias magnetic field)[9,48–50], and (ii) field-free deterministic SOT switching of perpendicular ferromagnetic layers[15,19,21,27]. In all cases, the CoTb samples with vertical composition gradient show the best SOT switching efficiency.

As expected in the SOT framework, the switching loops show opposite switching polarities under ±100 Oe. More interestingly, a full SOT switching loop in the absence of $H_{ex}$ is observed as well. This field-free switching loop has the same polarity as the cases with a positive $H_{ex}$. In addition, almost no switching signal is detected in this sample when $H_{ex} = -20$ Oe. These results indicate the existence of an effective internal field that participates in the SOT switching process. We attribute this effective field to the g-DMI in the CoTb layer[36].

To quantify the g-DMI-induced effective field $H_{DMI}$ in our sample, a method based on the magnetic droplet nucleation model was used[36,51]. As depicted in the inset of Fig. 3c, the hysteresis loop was measured by sweeping the magnetic field at an angle $\theta$ with respect to the $z$-axis. Figure 3c illustrates the magnetization switching curves in the negative field range with different angles. If we denote the magnetic field where the magnetization is switched from the "up" state to the "down" state by $H_{sw}$, then the coercive field $H_c$ and the accompanying in-plane field $H_n$ can be expressed by $H_{sw}cos\theta$ and $H_{sw}sin\theta$, respectively.

Figure 3d shows the measured $H_c$ as a function of $H_n$. As described in earlier works, the curve shows a clear plateau, and $H_c$ starts to decrease after $H_n$ passes a threshold value. This threshold $H_n$ corresponds to $H_{DMI}$[36,51]. In this manner, $H_{DMI}$ was determined to be around 23 Oe in the 6 nm CoTb layer with $\delta = 0.07$. We note that this value is very close to the absolute

value of applied in-plane $H_{ex}$ (−20 Oe, see Fig. 3b) where magnetization switching vanishes, which is consistent with the hypothesis that the g-DMI-induced effective field is responsible for the observed field-free switching[33–35].

To investigate the role of the composition gradient on the switching behavior in more detail, SOT switching experiments were also performed in the following samples: (A) 6 nm thick CoTb with $\delta = 0.06$ per 1 nm; (B) 9 nm thick CoTb with $\delta = 0.07$ per 1.5 nm; (C) 4.2 nm thick CoTb with $\delta = 0.07$ per 0.7 nm. Note that samples A and B have a smaller slope of the composition gradient compared to the previously discussed samples (where we had a 6 nm thick CoTb film with $\delta = 0.07$ per 1 nm), due to the smaller value of $\delta$ and the larger overall CoTb thickness, respectively. On the other hand, the slope of the composition gradient is increased in sample C, due to the reduced overall CoTb thickness for the same value of $\delta$.

The results are shown in Supplementary Notes S3, S4, and S5. All the samples exhibited clear deterministic switching loops in the absence of $H_{ex}$, and the value of $H_{ex}$ where switching vanished corresponded well to the measured value of $H_{DMI}$. This is in agreement with the fact that the g-DMI is responsible for the in-plane symmetry-breaking and zero-field switching observed in these samples. As expected, the $j_c$ values of samples A and B ($>10 \times 10^{10}$ A/m$^2$) were larger than that in the 6 nm thick CoTb layer with $\delta = 0.07$. On the contrary, a smaller $j_c$ ($\sim 7.5 \times 10^{10}$ A/m$^2$), as well as a higher $H_{DMI}$ ($\sim 40$ Oe), were achieved in sample C, as expected based on its steeper composition gradient.

In addition, we also studied the role of device orientation, device location on the wafer, and applied current history in the field-free SOT switching. These experiments are described in Supplementary Notes S9 and S10. They show that device orientation does not influence the field-free switching polarity, confirming that the switching is independent of conventional lateral asymmetry. Moreover, we found that the applied writing

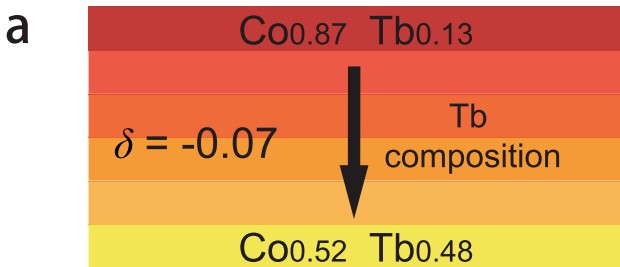

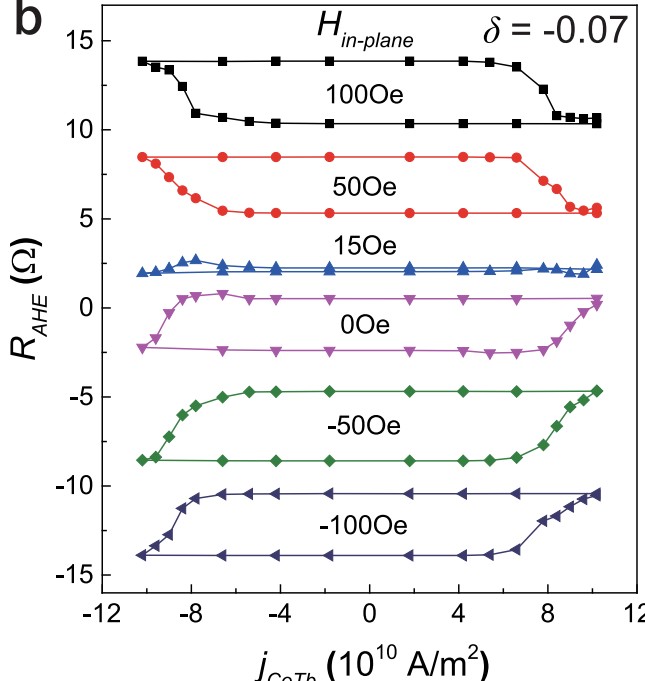

**Fig. 4 Comparison experiments on a CoTb sample with inverse gradient.**
**a** Structure of the sample with $\delta = -0.07$. **b** SOT switching curves of the
sample ($\delta = -0.07$) under different in-plane magnetic fields (including
zero-field). This sample shows opposite switching polarity as well as an
opposite DMI-induced field compared to the sample in Fig. 3b ($\delta = 0.07$),
verifying the vertical gradient-induced origin of SOT and DMI in our
samples.

current history provides a training effect that determines the
polarity of its subsequent current-induced switching loops. This
behavior was consistent for multiple devices measured across
different locations on a 100 mm wafer.

**Switching behavior of a CoTb film with an inverse composition
gradient**. To further verify the origin of SOTs and g-DMI, we
studied the SOT switching behavior in a 6 nm thick CoTb layer
with an inverse composition gradient ($\delta = -0.07$), i.e., Tb con-
centration decreasing from bottom to the top of the layers. The
details of this sample are described in Fig. 4a, while Fig. 4b shows
the obtained SOT switching curves of this sample under different
$H_{ex}$. The switching loops always show opposite polarity compared
to those of the sample with $\delta = 0.07$ (comparison of Figs. 4b
and 3b). This indicates that the vertical broken symmetry direc-
tions in these two samples are opposite, confirming again that the
SOTs in the CoTb layer truly originates from the vertical com-
position gradient.

SOT switching in the absence of $H_{ex}$ was also observed in this
sample, and the zero-field loop exhibits a polarity similar to

switching loops under negative $H_{ex}$. In addition, the deterministic
switching vanishes at a positive $H_{ex}$ of ~15 Oe. Both of these
characteristics, which are determined by g-DMI in our field-
free switching explanation, are opposite to the ones observed in
the sample with $\delta = 0.07$. Thus, we conclude that the inverse
composition gradient also induces an $H_{DMI}$ with opposite sign
compared to the previous case. The combination of the reversed
sign of SOT compared to our reference device, along with the
opposite sign of $H_{DMI}$ explains the field-free switching data of
Fig. 4b and is consistent with the observations in Fig. 3b.

**Modeling and simulations**. To shed light on the underlying
mechanism of this field-free switching phenomenon, we per-
formed micromagnetic simulations. Our micromagnetic frame-
work is based on a two-sublattice model strongly coupled through
an exchange interaction[41,52–56]. Along with the exchange inter-
action, we also take into account the anisotropy, the DMI, and the
torque exerted by SOT. Supplementary Note S6 provides details
of the micromagnetic model. The g-DMI originates from the
composition gradient along the **z** axis[36] and therefore the sym-
metry breaking is, from a modeling point of view, similar to the
case of interfacial DMI observed in other multilayer systems[57–59].

To mimic the experimental setup (see Fig. 2a) we consider a
cross geometry initially saturated along the $+z$ direction. From
this initial state, we apply an electric current for 20 ns (from $t_{on,1}$
to $t_{off,1}$), let the system relax for 12 ns more, and repeat the
process with the opposite polarization of the current (from $t_{on,2}$ to
$t_{off,2}$). Figure 5b shows the time evolution of the average $z$
component of the magnetization for the first sublattice (black
line) together with the timing of the two current pulses.

As observed in previous numerical studies[33–35,53], the
deterministic switching starts with the nucleation of a domain
at the edge of the sample because of the magnetization tilting
toward an in-plane direction imposed by the DMI boundary
conditions (see Fig. 5a). In other words, the nucleation of a new
domain at the edge is driven by the SOT applied to this in-plane
component at the edge (Fig. 5c–e). The switching occurs via a
strongly nonuniform magnetization pattern, given by the higher
DW velocity in the edges as compared to the central region due to
the combined effects of the DMI boundary conditions, the DMI,
and exchange fields (Fig. 5f–h). After the current is switched off,
the magnetization relaxes to the reversed uniform ground state
(Fig. 5i–j). The same process is observed for the current pulses of
opposite polarity (see Fig. S7 in Supplementary Note S7). To
consider a more realistic memory device geometry, we also
implemented similar simulations for a pillar with a diameter of
400 nm (see Fig. S8 in Supplementary Note S7). The main
conclusions about the switching process are not affected when
considering this pillar geometry.

**Discussion**
We have demonstrated a field-free DMI-SOT switching scheme
in a thick perpendicular CoTb layer with a vertical composition
gradient. The broken structural inversion symmetry along the
growth direction induces both strong SOTs and g-DMI. Notably,
this gradient-driven switching mechanism is promising to realize
field-free switching under a relatively low current density and has
the potential to be scalable to a large wafer size. The SOT
switching efficiency is significantly better than in previous field-
free SOT switching experiments in ferromagnets, as well as pre-
vious (field-assisted) SOT switching experiments in conventional
CoTb layers without composition gradient. Given that the SOT in
our structure originates within the bulk of the CoTb gradient film,
it could in principle be further enhanced by interfacing with an
appropriate heavy metal layer to provide additional interfacial

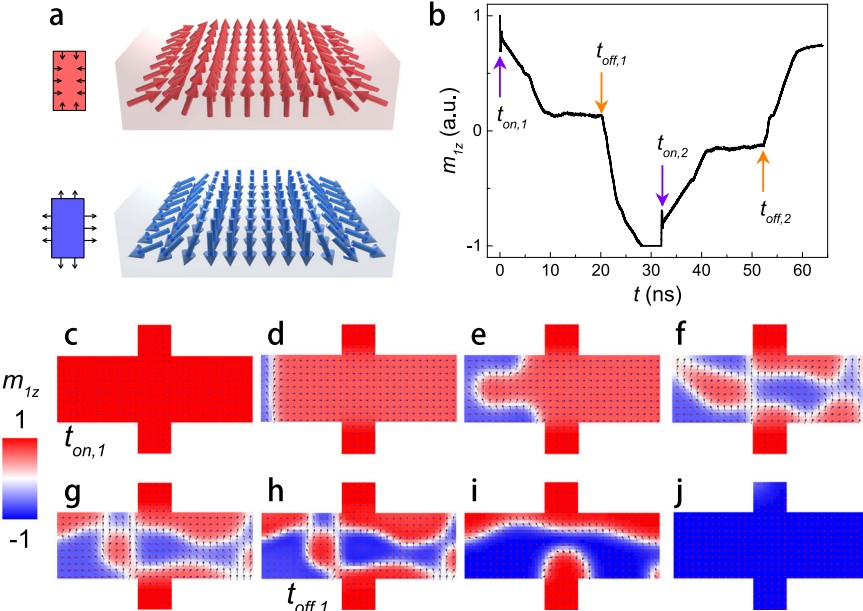

**Fig. 5 Micromagnetic simulations. a** A sketch of the magnetization tilting driven by DMI boundary conditions allowing domain nucleation at the edges. **b** Average z component of the first sublattice magnetization in the switching of the system from 0 ns to 32 ns, and from 32 ns to 64 ns, respectively (under opposite current directions). **c–j** First sublattice magnetization distribution at different times of the dynamics (0–32 ns).

SOT. Notably, since the g-DMI in our structure also originates from the composition gradient within the bulk, it does not impose any constraints on the choice of the heavy metal layer for interfacial SOT, thus providing freedom in device design. Finally, we note that large tunneling magnetoresistance (TMR) ratios have already been demonstrated in magnetic tunnel junctions based on FIMs[60–62]. Therefore, it is possible to integrate the CoTb gradient films presented here into SOT memory devices with TMR readout. Our findings go beyond the conventional paradigm of SOT-induced perpendicular magnetization switching with the assistance of external field and SOT sources and being applicable to ferrimagnets, pave the way towards the development of ultrafast SOT memory arrays.

## Methods

**Sample growth and characterization**. A series of Al (2)/CoTb (6)/Al (3) (thickness in nm) stacks were deposited on thermally oxidized silicon substrates by DC and RF sputtering under a base pressure lower than $10^{-8}$ Torr. The composition of CoTb was controlled by setting the deposition power of Co and Tb targets. Vibrating sample magnetometry was used to quantify the magnetic properties of each sample. STEM and EDS characterizations were performed by a JEM-ARM200F microscope.

**Device fabrication and electrical measurement**. The films were fabricated into Hall bar devices of 10 μm width by optical lithography and ion milling techniques. For AHE and DMI measurements, DC current was applied along the x axis, while for SOT switching measurements, current pulses (0.1 ms width) were applied. A Hall probe was inserted in the probe station to detect the actual magnetic field values during measurement, thus guaranteeing the absence of any $H_{ex}$ contributions due to remanent fields. For harmonics measurements, two SR830 lock-in amplifiers were used to detect the first and second harmonic Hall voltage induced by an AC current with a frequency of 133.33 Hz.

## Data availability

The data that support the findings of this study are available from the corresponding authors upon reasonable request.

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

## Acknowledgements

This work was supported by a grant from the U.S. National Science Foundation, Division of Electrical, Communications and Cyber Systems (NSF ECCS-1853879), the National Natural Science Foundation of China (Grant no. 61971024 and 51901008), the International Mobility Project (Grant no. B16001) and National Key Technology Program of China (Grant no. 2017ZX01032101). This work was also supported by the National Science Foundation Materials Research Science and Engineering Center at Northwestern University (NSF DMR-1720319) and made use of its Shared Facilities at the Northwestern University Materials Research Center. This work also made use of the NUFAB facility of Northwestern University's NUANCE Center, which has received support from the SHyNE Resource (NSF ECCS-1542205), the IIN, and Northwestern's MRSEC program (NSF DMR-1720139). Z.Y.Z also acknowledges the support from the China Scholarship Council (No. 201906020022). G.F., L.S.T, and M.C. would like to acknowledge the contribution of the COST Action CA17123 "Ultrafast optomagneto-electronics for nondissipative information technology".

## Author contributions

P.K.A., W.S.Z., and Y.Z. planned and supervised the project. Z.Y.Z. and Y.Z. designed the experiments and fabricated the devices with help from V.L.-D. and J.S. Z.Y.Z., V.L.-D., X.F., L.C., Z.W., Z.Z., and Y.X performed the measurements of the samples and analyzed the data. B.H., K.Z., and Y.G.Z. performed the STEM imaging. L.S.-T., M.C., and G.F. implemented the micromagnetic code and performed the simulations. A.F. contributed to the theoretical explanation and data interpretation. Z.Y.Z., P.K.A., G.F., and Y.Z. co-wrote the manuscript. All the authors read and commented on the manuscript.

## Competing interests

The authors declare no competing interests.
