## [Peer Review File · Nature Communications]

Reviewers' Comments:

Reviewer #1:

Remarks to the Author:

Zheng et al reported field-free spin-orbit torque switching (SOT) of a single ferrimagnetic layer with vertical composition gradient. The composition gradient induces a non-zero SOT and Dzyaloshinskii-Moriya interaction (DMI). However, it is presumably assumed that there is no lateral symmetry breaking in the layer. It has been well known (also well described in the introduction of the manuscript) that a lateral symmetry breaking is required for field-free SOT switching. Interestingly, the authors observed that the single ferrimagnetic layers without such lateral symmetry breaking show field-free SOT switching. An effective in-plane field ($= -20$ Oe, in Fig. 3c) is close to the DMI field so that the authors claimed that the DMI results in the additional symmetry breaking. This claim is in line with modeling results of Refs. [33-35] and their own modeling result.

This is an interesting result, but I could not understand why the field-free SOT switching is allowed even though there is no lateral symmetry breaking. Any physical phenomena must obey the symmetry rule regardless of microscopic details. Symmetry-wise, the investigated structure is the same as ferromagnet/heavy metal bilayers, which do not allow field-free switching unless the lateral symmetry is broken. The authors (and Refs. [33-35]) argued that the system is laterally symmetric in equilibrium, but the symmetry is broken during dynamics. I cannot agree with this argument. The authors must show a simple symmetry argument to explain their observation without relying on the numerical simulations. Without a clear symmetry argument (how the DMI breaks the lateral symmetry breaking), it is difficult to trust the modeling result because the numerical modeling in a system with DMI would be highly nonlinear and the DMI boundary condition for actually stair-case shaped edges would be problematic.

Overall I do not support publication of this manuscript in Nat. Comm. unless a simple and clear symmetry argument supporting the field-free SOT switching is provided.

Reviewer #2:

Remarks to the Author:

In this manuscript, the authors investigated spin-orbit torque (SOT) switching of perpendicular magnetizations in a ferrimagnetic layer with vertical composition gradient. Using the combination of SOT and gradient-driven Dzyaloshinskii-Moriya interaction (g-DMI), field-free magnetization switching in a single ferrimagnetic layer was demonstrated through experiments. The method used for in-plane symmetry breaking is novel. The study is timely and thus it is interesting to the spintronic research community. However, there are several points that still need to be clarified.

1. The field-free switching demonstrated in the manuscript is for a micron-sized Hall-bar structure. Is it possible for it to scale down to a nano-sized pillar structure? Please justify.
2. The switching mechanism is yet to be clearly understood. In the micromagnetic simulation, the authors assumed an initial reversed domain at the edge of the sample, which makes the field-free switching possible. They claimed that this is due to the magnetization tilting driven by DMI boundary conditions which allows the domain to nucleate at the edges. If this is true, then there is no need to assume an initial reversed domain at the edge since DMI is already included in the simulation. Are there any pieces of experimental evidence that the domain nucleates at the edges of their Hall-bar structure? What are the factors that may cause this to occur?
3. Since the experiment was conducted at finite temperature, has the temperature effect been considered in the simulation?
4. How are the simulation parameters in S6 obtained? Any experimental works from which those values are based on?
5. The authors discretized the system with tetragonal cells of $10 \times 10 \times 6$ nm³ in their simulation. What is the system size they were simulating?
6. In the second last line of Page 9, the equation reads $j_{\text{CoTb}} \partial \text{DL}_{\text{top}} = j_{\text{CoTb}} (\partial \text{DL}_{\text{top}} - \partial \text{DL}_{\text{bottom}})$. This equation seems not correct.
7. In Page 11, the authors write "If we denote the magnetic field where the magnetization is

switched from the “up” state to the “down” state by H_{sw} , then, the coercive field H_c and the accompanying in-plane field H_n are determined to be $H_{sw}\sin\theta$ and $H_{sw}\cos\theta$, respectively.” The definitions of H_c and H_n should be reversed, that is $H_c = H_{sw}\cos\theta$ and $H_n = H_{sw}\sin\theta$?
8. Equation S5 is not clearly shown.

Reviewer #3:

Remarks to the Author:

This is a very good paper which reports bias-field-free SOT switching in a single perpendicular CoTb layer with an engineered vertical composition gradient. The authors experimentally show that the vertical structural inversion asymmetry induces SOTs and a gradient-driven Dzyaloshinskii–Moriya interaction (g-DMI). The g-DMI breaks the in-plane symmetry and makes the bias-field-free SOT switching possible. Especially, they show that the sign of g-DMI depends on the sign of vertical composition gradient, which is a strong proof that the observed DMI originates from the composition gradient.

The paper is well written and the experimental result is original and clear. I recommend the publication of this paper in Nature Communications after revisions.

My only concern is the initial magnetic state in the micromagnetic simulation. There is a domain wall at the edge, and the magnetization reversal proceeds by the motion of this domain wall. What happens if the simulation starts from a fully saturated state without a domain wall?

Regarding this concern, I would like to request the authors to add the simulation for circular shape structure which is more relevant to a real device structure than the present device shape in the manuscript.

Response Letter

COMMENTS TO AUTHOR:

Reviewer #1:

Zheng et al reported field-free spin-orbit torque switching (SOT) of a single ferrimagnetic layer with vertical composition gradient. The composition gradient induces a non-zero SOT and Dzyaloshinskii-Moriya interaction (DMI). However, it is presumably assumed that there is no lateral symmetry breaking in the layer. It has been well known (also well described in the introduction of the manuscript) that a lateral symmetry breaking is required for field-free SOT switching. Interestingly, the authors observed that the single ferrimagnetic layers without such lateral symmetry breaking show field-free SOT switching. An effective in-plane field ($= -20$ Oe, in Fig. 3c) is close to the DMI field so that the authors claimed that the DMI results in the additional symmetry breaking. This claim is in line with modeling results of Refs. [33-35] and their own modeling result.

This is an interesting result, but I could not understand why the field-free SOT switching is allowed even though there is no lateral symmetry breaking. Any physical phenomena must obey the symmetry rule regardless of microscopic details. Symmetry-wise, the investigated structure is the same as ferromagnet/heavy metal

bilayers, which do not allow field-free switching unless the lateral symmetry is broken. The authors (and Refs. [33-35]) argued that the system is laterally symmetric in equilibrium, but the symmetry is broken during dynamics. I cannot agree with this argument. The authors must show a simple symmetry argument to explain their observation without relying on the numerical simulations. Without a clear symmetry argument (how the DMI breaks the lateral symmetry breaking), it is difficult to trust the modeling result because the numerical modeling in a system with DMI would be highly nonlinear and the DMI boundary condition for actually stair-case shaped edges would be problematic.

Overall I do not support publication of this manuscript in Nat. Comm. unless a simple and clear symmetry argument supporting the field-free SOT switching is provided.

Authors' Response:

Thank you very much for this insightful comment. We performed additional experiments to further clarify the nature of the observed deterministic switching, and correspondingly, to determine the symmetry requirements that would need to be met, to be consistent with the experimental data. These new results and the associated discussion of the experiment's symmetry are discussed below. We hope that this analysis and the related updates to the revised manuscript will address your concern.

Summary of the new experimental results and their symmetry properties:

Before presenting the details of the new results, we would like to summarize the main findings and conclusions:

1. We performed additional measurements on a series of newly constructed devices, all of which consistently exhibited the same type of switching reported in the original manuscript. Notably, we confirmed that the switching is both deterministic and directional in all of these samples, i.e. that a particular direction of current favors a particular direction of out-of-plane magnetization.
2. We further performed this same experiment on a series of devices on the same

wafer, made at different angles with respect to the initial device axis. All of these devices exhibited the same directionality in the current-induced switching, regardless of the device angle. Collectively, these two experiments provide strong evidence that (i) the deterministic and directional switching is robust and reproducible, and (ii) that the directionality of the switching is not a result of possible unintentional asymmetries in the device structure (e.g. due to the fabrication process), since such unintentional asymmetries would not show the same directionality of the switching across a large number of devices.

3. The key remaining question, then, is what associates a particular direction of in-plane current with a particular direction of out-of-plane magnetization, if there is no structural asymmetry in the device? Our experiments, as described below, showed that the *direction of the initial current pulse sent through the device* (i.e. the *training* during the first measurement of a current-induced switching) determines this directionality. This *training* effect itself breaks the symmetry of the experiment. In other words, no symmetry requirements are violated when the history of the first applied current is considered in conjunction with the directionality of the loops in our devices, so that different directions of the training pulse are consistently associated with different directionalities of the subsequent current-induced switching loops.

In what follows, we first describe the experiments that support the above-stated conclusions, followed by a qualitative discussion of the possible micromagnetic explanation of these results.

Details of new experimental results:

We fabricated new samples and performed a series of additional experiments which are described below. The first question we sought to answer was to verify that the field-free SOT-induced switching in our devices is indeed directional, i.e. that each current direction favors a particular final state (up or down), and that this directionality is consistent across a large number of devices.

To do so, we first performed a time-domain switching measurement, as illustrated in Fig. R1. The purpose of this experiment was to illustrate that the final state of the

device after switching is determined by the direction of the writing current, but not by the number of consecutive current pulses sent in the same direction, i.e. that the device is not undergoing toggle switching.

The results can be seen in Fig. R1, where black pulses indicate write attempts in different directions (0.1 ms , $+11 \text{ MA/cm}^2$ or -11 MA/cm^2) and red circles correspond to the subsequent readout, i.e. the measurement of the R_{AHE} of CoTb. Note that we performed two consecutive read attempts in each case, to also verify that the readout process itself, i.e. the measurement of R_{AHE} , does not change the state of the device in any way.

It is clearly observed from Fig. R1 that the first writing pulse in each direction switches the magnetization of the CoTb in the corresponding up/down direction, but that subsequent pulses in the same direction (regardless of their number, i.e. whether one, two, or three consecutive pulses are used) do nothing to change the state of the bit. This clearly shows that the applied current pulse polarity (but not the number of pulses in the same direction) changes the R_{AHE} value, indicating that the switching is deterministic and directional.

Note that, this behavior is qualitatively different from what Refs. [1.1-1.3] (Refs. [33-35] in the main text) predicted from micromagnetic simulations. In those simulations, the writing current, no matter what direction it had, always flipped the bit to the opposite magnetic state, thus exhibiting toggle switching.

Fig. R1 Time-domain measurement with writing pulses in both directions, demonstrating that the switching is deterministic and directional.

Next, we performed additional experiments to verify whether any (unintentional)

in-plane asymmetry may play a role in our observations. To do so, a direct evidence would be whether the in-plane device orientation impacts the switching polarity. To answer this question, we fabricated new samples where devices with different angles are arrayed on the same chip (see Fig. R2).

Fig. R2 Optical image of the fabricated sample where devices with different angles are arrayed.

As illustrated in Fig. R3 (left panel), we first measured the SOT switching loops in four different devices with 0 degree, 45 degree, 90 degree and 135 degree orientation configurations, respectively. It is clearly seen that all of them exhibited the same switching polarity. Next, we rotated the measurement setup by 180 degree in these four devices to measure the switching loops again, corresponding to the 180 degree, 225 degree, 270 degree and 315 degree orientation configurations, as indicated in the right panel of Fig. R3. As would be expected from a directional switching behavior, the switching loops were all inverted in this case.

To verify that this inverted switching polarity is repeatable in a single device, we randomly chose a 45 degree device on the chip and measured it with both configurations, i.e. the 45 degree and 225 degree setups shown in Fig. R3, for several times. As shown in Fig. R4, the directionality of the switching can be clearly repeated, and therefore cannot be attributed to any random event.

It cannot be emphasized enough, that the results of Fig. R3 indeed seem to defy the symmetry of the structure. However, our next experiment showed that this is not the case, and in fact all symmetry considerations are indeed satisfied when the experiment is considered in its entirety, including the training effect from the initial current pulse applied to the device.

The only possible symmetry-breaking element between the first four

measurements (left panel in Fig. R3) and the second four measurements (right panel in Fig. R3) is that in the second four measurements, current pulses have already been injected in the device before, i.e., the devices have been trained by the first current pulse which has been applied in the positive direction of the device, as defined on the left panel of Fig. R3.

To control for this training variable, we chose several new devices, in which no current had been applied before subsequent to device fabrication. Fig. R5 shows the corresponding SOT switching loops for three of these devices. We notice that in sharp contrast to Fig. R3, in this case all the devices (0 degree, 180 degree and 225 degree) show the same SOT-induced switching directionality. Notably, the cases of 180 degree and 225 degree devices show the opposite directionality compared to Fig. R3. This can be explained based on the training effect provided by the initial current sent through the devices, which is opposite for these two angles in Figs. R3 and R5, and therefore breaks the symmetry of the experiment.

Fig. R3 SOT switching loops of devices varying from 0 degree to 135 degree and the corresponding loop of each device with a 180 degree inverted measurement setup.

Fig. R4 SOT switching experiment repetitions with a 180 degree rotated setup.

Fig. R5 SOT switching loops of new devices (with no prior training) under different angles: (a) 0 degree, (b) 180 degree and (c) 225 degree.

Discussion and possible micromagnetic interpretation:

Given the size of our devices, it is expected that SOT-induced magnetization switching is completed by domain wall (DW) motion, as is the case in most other SOT-induced switching experiments reported to date. Inspired by this DW-mediated switching character, recently a field-free deterministic switching scheme has been proposed that relies on geometrical pinning of DWs [1.4]. Similar to our work, the device structure proposed in Ref. [1.4] also does not break the lateral in-plane *structural* symmetry; however, its deterministic switching is allowed because of the broken inversion symmetry of the *micromagnetic* structure of the device.

We hypothesize that in our devices, the first current pulse applied to the device creates a chiral magnetic texture due to the simultaneous effect of SOT and g-DMI. For example, this could be in the form of the nucleation of initial DWs with a chirality

that depends on the g-DMI, at the edges of the device. These DWs are possibly pinned by defects and maintain their existence during magnetization switching. Similar to Ref. [1.4], their presence can break the micromagnetic in-plane symmetry of the experiment and in principle allows for deterministic switching to occur. The experimental data as a function of the in-plane field support this interpretation.

The chirality of these DWs is fixed by a competition between the g-DMI and the external in-plane field. In fact, it can be noted that reversed loops are observed for an in-plane field larger than the g-DMI field. When the two are comparable (e.g. -20 Oe in Fig. 3b and 15 Oe in Fig. 4b), the current does not bring about a deterministic change in the resistance because of the Bloch type DWs stabilized by the magnetostatic field.

Stated differently, the creation of these chiral textures at the edge of the device is a local symmetry-breaking event related to g-DMI and SOTs. Because of it, deterministic and directional field-free SOT-induced switching is in principle allowed by symmetry, even though it cannot be explained by the global symmetry of the device structure.

A remaining question, of course, is the reconfigurability of the observed training effect, i.e. whether the devices could possibly be *re-trained* to exhibit an opposite directionality of the switching loops, e.g. by driving a larger current through them, opposite to the initial training current pulse. We were not able to observe such a re-training effect in our devices, which may be because of the practical limit on the maximum current amplitude that we could apply before breaking the devices.

While the simulation in Fig. 5 of the revised main text explains the detailed switching process, we note and appreciate your concern over the implementation of the DMI boundary condition for stair-case shaped edges in numerical modeling. We claim that this procedure is in fact not problematic, since it has been implemented by introducing a shape function which properly identifies the boundary between the ferromagnetic and non-ferromagnetic regions. As stated in Eq. (S6), the boundary conditions for the first sublattice are

$$2A_{11}\partial_n\vec{m}_1 + D\vec{m}_1 \times (\vec{n} \times \vec{u}_z) + A_{12}\vec{m}_1 \times (\partial_n\vec{m}_2 \times \vec{m}_1) = \vec{0}. \quad (\text{R1})$$

A similar expression applies for the second sublattice. The numerical implementation of the boundary condition for the right edge is then

$$\begin{cases} m(N_x + 1, c_y, c_z)_{i,x} = m(N_x, c_y, c_z)_{i,x} - \xi \left(m(N_x, c_y, c_z)_{i,z} - \frac{A_{12}}{2A_{11}} m(N_x, c_y, c_z)_{j,z} \right) \\ m(N_x + 1, c_y, c_z)_{i,y} = m(N_x, c_y, c_z)_{i,y} \\ m(N_x + 1, c_y, c_z)_{i,z} = m(N_x, c_y, c_z)_{i,z} + \xi \left(m(N_x, c_y, c_z)_{i,x} - \frac{A_{12}}{2A_{11}} m(N_x, c_y, c_z)_{j,x} \right) \end{cases}, \quad (\text{R2})$$

where the pair of parameters i and j can have the following values, $i, j = (1,2)$ or $i, j = (2,1)$, and $\xi = \frac{D\Delta x}{2A_{11}\left(1 - \left(\frac{A_{12}}{2A_{11}}\right)^2\right)}$.

For the special case where $A_{12} = -2A_{11}$, we have the following equations for the x component and x edge,

$$\begin{cases} 2A_{11}(\partial_x m_{1,x} - \partial_x m_{2,x}) = -Dm_{1,z} \\ 2A_{11}(\partial_x m_{2,x} - \partial_x m_{1,x}) = -Dm_{2,z} \end{cases}, \quad (\text{R3})$$

from which we obtain the equation: $m_{1,z} = -m_{2,z}$. Using this condition, one can rewrite the general case for the x edges and the x component as

$$\partial_x m_{1,x} = -\frac{D}{2A_{11}\left(1 - \left(\frac{A_{12}}{2A_{11}}\right)^2\right)} \left(m_{1,z} - \frac{A_{12}}{2A_{11}} m_{2,z} \right) = -\frac{D}{2A_{11}\left(1 - \left(\frac{A_{12}}{2A_{11}}\right)^2\right)} \left(1 + \frac{A_{12}}{2A_{11}} \right) m_{1,z}, \quad (\text{R4})$$

which finally transforms into

$$\partial_x m_{1,x} = -\frac{D(2A_{11}+A_{12})m_{1,z}}{(2A_{11}-A_{12})(2A_{11}+A_{12})} = -\frac{Dm_{1,z}}{(2A_{11}-A_{12})} = -\frac{Dm_{1,z}}{4A_{11}}. \quad (\text{R5})$$

By introducing the above-mentioned boundary condition, a current pulse, in combination with the chiral symmetry breaking brought about by g-DMI, will create a DW at the edge (Figs. 5c-d in the main text), so that subsequently the magnetization switching can be completed via propagation of DWs (Figs. 5e-j in the main text).

Here, we would like to sincerely thank you again for your insightful comments that allowed us to carefully examine and better understand our experimental data. Below is the summary of the modifications on this point in the revised manuscript.

Indication of changes in the revised manuscript:

- 1) We added the new experimental results as Supplementary Note S9.
- 2) Main text: Page 13. We added ‘‘In addition, we also studied the role of device orientation and applied current history in the field-free SOT switching. These

experiments are described in Supplementary Note S9. They show that device orientation does not influence the field-free switching polarity, confirming that the switching is independent of conventional lateral asymmetry. Moreover, we found that the applied writing current history provides a training effect that determines the polarity of its subsequent current-induced switching loops.”

References

- [1.1]Chen, B., Lourembam, J., Goolaup, S. & Lim, S. T. Field-free spin-orbit torque switching of a perpendicular ferromagnet with Dzyaloshinskii-Moriya interaction. *Appl. Phys. Lett.* **114**, 022401 (2019).
- [1.2]Wu, K., Su, D., Saha, R. & Wang, J. P. Deterministic field-free switching of a perpendicularly magnetized ferromagnetic layer via the joint effects of the Dzyaloshinskii–Moriya interaction and damping-and field-like spin–orbit torques: an appraisal. *J. Phys. D Appl. Phys.* **53**, 205002 (2020).
- [1.3]Dai, M. & Hu, J. M. Field-free spin–orbit torque perpendicular magnetization switching in ultrathin nanostructures. *NPJ Comput. Mater.* **6**, 1-10 (2020).
- [1.4]Lee, J. M., et al. Field-free spin–orbit torque switching from geometrical domain-wall pinning. *Nano Lett.* **18**, 4669-4674 (2018).

Reviewer #2: In this manuscript, the authors investigated spin-orbit torque (SOT) switching of perpendicular magnetizations in a ferrimagnetic layer with vertical composition gradient. Using the combination of SOT and gradient-driven Dzyaloshinskii–Moriya interaction (g-DMI), field-free magnetization switching in a single ferrimagnetic layer was demonstrated through experiments. The method used for in-plane symmetry breaking is novel. The study is timely and thus it is interesting to the spintronic research community. However, there are several points that still need to be clarified.

Authors' Response:

We greatly appreciate your positive assessment of this work, and your constructive suggestions which helped us improve the quality of this manuscript.

1. The field-free switching demonstrated in the manuscript is for a micron-sized Hall-bar structure. Is it possible for it to scale down to a nano-sized pillar structure? Please justify.

Authors' Response:

The switching is not shape-dependent, thus, we believe that scaling down to nano-sized pillars should not hinder the field-free switching. We have added the information regarding the simulated dynamics on a circular shape (400 nm diameter) device in the Supplementary Note S7, which is also reproduced below for the sake of clarity. Please see the following Fig. R6 for the detailed switching dynamics. The writing current pulse is applied from 0 ns to 20 ns. Thus, our conclusions are not affected when considering a pillar geometry.

Fig. R6 Micromagnetic simulations in a pillar with 400 nm diameter. Panels a-j show the first sublattice magnetization distribution at different times of the switching dynamics in the pillar, confirming that a switching process similar to that of the main text also occurs in this geometry.

Indication of changes in the revised manuscript:

- 1) We added the new simulation of the pillar geometry in Supplementary Note S7.
 - 2) Main text: Page 15. “To consider a more realistic memory device geometry, we also implemented similar simulations for a circular pillar with a diameter of 400 nm (See Fig. S8 in Supplementary Note S7). The main conclusions about the switching process are not affected when considering this pillar geometry.”
2. The switching mechanism is yet to be clearly understood. In the micromagnetic simulation, the authors assumed an initial reversed domain at the edge of the sample, which makes the field-free switching possible. They claimed that this is due to the magnetization tilting driven by DMI boundary conditions which allows the domain to nucleate at the edges. If this is true, then there is no need to assume an initial reversed domain at the edge since DMI is already included in the simulation. Are there any pieces of experimental evidence that the domain nucleates at the edges of their Hall-bar structure? What are the factors that may cause this to occur?

Authors' Response:

Thank you for noting this point. Indeed, the initial state in our simulation is the fully saturated state without any domain at the edge. However, we showed a state where the domain wall has already been nucleated (at $t=0.63$ ns) as the first snapshot of the dynamics, leading to this misunderstanding. We have updated the figure including the initial state to avoid any confusion. The following Figs. R7 and R8 illustrate this, and correspond to Fig. 5 in the main text and Fig. S7 in Supplementary Note S7, respectively.

Fig. R7 Micromagnetic simulations. a, A sketch of the magnetization tilting driven by DMI boundary conditions allowing domain nucleation at the edges. b, Average z component of the first sublattice magnetization in the switching of the system from 0 ns to 32 ns, and from 32 ns to 64 ns, respectively (under opposite current directions). c-j, First sublattice magnetization distribution at different times of the dynamics (from 0 ns to 32 ns).

Fig. R8 Micromagnetic simulations in a Hall bar. a-h, First sublattice magnetization distribution at different times of the dynamics (from 32 ns to 64 ns).

Indication of changes in the revised manuscript:

We updated the above-mentioned figures, which correspond to Fig. 5 in the main text and Fig. S7 in Supplementary Note S7.

3. Since the experiment was conducted at finite temperature, has the temperature effect been considered in the simulation?

Authors' Response:

In the simulations presented in the manuscript and the Supplementary Notes, we have taken into account the thermal effects considering $T=300$ K. Without thermal effects the current density needed for the switching would be larger. We expanded the discussion about the numerical implementation of thermal fields in the Supplementary Note S6. Additionally, the effective field is augmented with a Gaussian random field to mimic the effect of the temperature. Thus, $\mathbf{H}_{eff,i}$ in Eq. (S2) in Supplementary Note S6 is substituted by $\mathbf{H}_{eff,i} + \mathbf{H}_{th,i}$, where

$$\mathbf{H}_{th,i} = \boldsymbol{\eta} \sqrt{\frac{2\alpha(1+\alpha^2 k_B T)}{\gamma_0 \mu_0 M_s V dt}}, \quad (\text{R6})$$

with k_B , T and V being the Boltzmann constant, the temperature (set to 300 K in our case) and the computational cell volume, respectively. In this equation, $\boldsymbol{\eta}$ is a vector

whose Cartesian components are random numbers following the Gaussian distribution:

$$\begin{cases} \langle \eta_k(t) \rangle = 0 \\ \langle \eta_k(t) \eta_l(t') \rangle = \delta_{kl} \delta(\mathbf{r} - \mathbf{r}') \delta(t - t') \end{cases} \quad (\text{R7})$$

Indication of changes in the revised manuscript:

We added the above description of the thermal effects in Supplementary Note S6.

4. How are the simulation parameters in S6 obtained? Any experimental works from which those values are based on?

Authors' Response:

Most of the simulation parameters are derived from the experiments in this work or from other experimental works on similar material systems. You can find the table below summarizing how the parameters have been identified.

Parameter	Value	Source
M_{S1}	654 kA/m	This work (see Fig. 1d). We assumed Co magnetization to be independent of Tb composition.
M_{S2}	609 kA/m	This work (see Fig. 1d). We assumed Tb magnetization to be independent of Co composition.
$K_{u1}; K_{u2}$	20 kJ/m ³	This work (see Table S1 in Supplementary Note S8), assuming $K_{u1}=K_{u2}$. Note that the effective value is the sum.
D	16 $\mu\text{J}/\text{m}^2$	This work (see Fig. 3).
A_{11}	3 pJ/m	Assumed
A_{12}	-3 pJ/m	Assumed
A_0	-6 pJ/m	Assumed
θ_{SH}	0.06	This work (see Fig. 2d).
γ_1	2.42×10^5 m/As	Literature ref [2.1-2.2].
γ_2	1.69×10^5 m/As	Literature ref [2.1-2.2].
α	0.05	Assumed

Indication of changes in the revised manuscript:

We added these detailed parameters in Supplementary Note S6.

5. The authors discretized the system with tetragonal cells of $10 \times 10 \times 6 \text{ nm}^3$ in their simulation. What is the system size they were simulating?

Authors' Response:

As shown in the following Fig. R9, the arm width is 200 nm, the arm length is 600 nm, and the thickness is 6 nm. We used a discretization grid of $60 \times 40 \times 1$ cells.

Fig. R9 Geometrical information of the simulated device.

Indication of changes in the revised manuscript:

We included the geometrical details in the Supplementary Note S6 (Fig. S6).

6. In the second last line of Page 9, the equation reads $j_{\text{CoTb}} \vartheta_{\text{DL,top}} = j_{\text{CoTb}} (\vartheta_{\text{DL,top}} - \vartheta_{\text{DL,bottom}})$. This equation seems not correct.

Authors' Response:

Thank for pointing out this typo. In fact, the correct equation should be $j_{\text{CoTb}} \theta_{\text{DL}} = j_{\text{CoTb}} (\theta_{\text{DL,top}} - \theta_{\text{DL,bottom}})$.

Indication of changes in the revised manuscript:

To clarify the meaning of this equation, we have done some modifications on the corresponding paragraph (Main text, Page 9), as follows: “In contrast, the asymmetric distribution of CoTb layers of different compositions, with higher 5d concentrations and larger Rashba spin polarizations at top, can give rise to unbalanced spin transfers

between different parts of the sample. In our samples, there should be some imbalance between the transfer to bottom magnetization of the larger downward spin currents $j_{CoTb}\theta_{DL,top}$ emitted from top and the transfer to top of the smaller upward spin currents $j_{CoTb}\theta_{DL,bottom}$ emitted with opposite spin polarization from bottom. The final torque is difficult to predict precisely because it will depend on how the gradient of Tb concentration is reflected in the variation of both the Rashba interaction and other parameters, such as the spin dephasing length, controlling the proportion of spin transfer. In rough approximation, the DL torque should correspond to the transfer of a spin current of the order of $j_{CoTb}(\theta_{DL,top} - \theta_{DL,bottom})$ (the bracket corresponds to an effective spin Hall angle θ_{DL} which, as expected, is zero if the layer is homogeneous).”

7. In Page 11, the authors write “If we denote the magnetic field where the magnetization is switched from the “up” state to the “down” state by H_{sw} , then, the coercive field H_c and the accompanying in-plane field H_n are determined to be $H_{sw}\sin\theta$ and $H_{sw}\cos\theta$, respectively.” The definitions of H_c and H_n should be reversed, that is $H_c = H_{sw}\cos\theta$ and $H_n = H_{sw}\sin\theta$?

Authors’ Response:

Thank for pointing out this typo.

Indication of changes in the revised manuscript:

We corrected this mistake in the revised main text, Page 11.

8. Equation S5 is not clearly shown.

Authors’ Response:

Thank you for pointing this out. We have double-checked in the new version to make sure this equation shows correctly.

Indication of changes in the revised manuscript:

We corrected this mistake in Supplementary Note S6.

References

- [2.1] Siddiqui, S. et al. Current-induced domain wall motion in a compensated ferrimagnet. *Phys. Rev. Lett.* **121**, 057701 (2018).
- [2.2] Finley, J. & Liu, L. Spin-orbit-torque efficiency in compensated ferrimagnetic cobalt-terbium alloys. *Phys.Rev. Appl.* **6**, 054001 (2016).

Reviewer #3: This is a very good paper which reports bias-field-free SOT switching in a single perpendicular CoTb layer with an engineered vertical composition gradient. The authors experimentally show that the vertical structural inversion asymmetry induces SOTs and a gradient-driven Dzyaloshinskii–Moriya interaction (g-DMI). The g-DMI breaks the in-plane symmetry and makes the bias-field-free SOT switching possible. Especially, they show that the sign of g-DMI depends on the sign of vertical composition gradient, which is a strong proof that the observed DMI originates from the composition gradient.

The paper is well written and the experimental result is original and clear. I recommend the publication of this paper in Nature Communications after revisions.

Authors' Response:

Thank you very much for the recognition of our work and for recommending publication of this manuscript in *Nature Communications*.

My only concern is the initial magnetic state in the micromagnetic simulation. There is a domain wall at the edge, and the magnetization reversal proceeds by the motion of this domain wall. What happens if the simulation starts from a fully saturated state without a domain wall?

Regarding this concern, I would like to request the authors to add the simulation for circular shape structure which is more relevant to a real device structure than the present device shape in the manuscript.

Authors' Response:

Thank you for this helpful suggestion. We would like to clarify that the initial state in our simulation is indeed the fully saturated state without any domain at the edge. However, in the previous version of the manuscript, we showed a state where the domain wall had already been nucleated (at $t=0.63$ ns) as a first snapshot of the dynamics, leading to this misunderstanding. Indeed, in our simulations, the switching

can be completed without a domain wall at the edge, which is updated in the new version of Fig. 5 in the main text.

Secondly, we have also added the information regarding the simulated dynamics on a pillar of 400 nm diameter in the Supplementary Note S7. Please see Fig. R6 for the detailed switching dynamics. The writing current pulse is applied from 0 ns to 20 ns. Thus, our conclusions are not affected when considering a pillar geometry.

Fig. R6 Micromagnetic simulations in a pillar with 400 nm diameter. Panels a-j show the first sublattice magnetization distribution at different times of the switching dynamics in the pillar, confirming that a switching process similar to that of the main text also occurs in this geometry.

Indication of changes in the revised manuscript:

- 1) We added the new simulation of the pillar geometry in Supplementary Note S7.
- 2) Main text: Page 15. “To consider a more realistic memory device geometry, we also implemented similar simulations for a pillar with a diameter of 400 nm (See Fig. S8 in Supplementary Note S7). The main conclusions about the switching process are not affected when considering this pillar geometry.”
- 3) We updated the simulation of the Hall bar without any initial domain walls at the edge in Fig. 5 in the main text and Fig. S7 in Supplementary Note 7.

Other modifications:

- 1) We noticed that Ref. [39] in the main text has been recently published in *Advanced Materials*. Thus, the citation has been changed from the arXiv reference to a journal citation.
- 2) We added two more funding sources in the Acknowledgements section.

Reviewers' Comments:

Reviewer #1:

Remarks to the Author:

The revised manuscript includes additional switching experiments showing that the first current pulse (e.g., its polarity) has a definite role in the symmetry breaking and field-free switching. However, a concrete description about the symmetry breaking mechanism associated with the "first current pulse" is still ambiguous. The authors speculate that a creation of a chiral magnetic texture due to the simultaneous effect of SOT and g-DMI breaks the symmetry. To my view, a speculation about the underlying mechanism is not sufficient to be publishable in a high profile journal like Nature Communications.

Q1. Is it not possible to experimentally prove the creation of a chiral magnetic texture to break the symmetry?

Q2. Why is the Co/Tb special for this speculated symmetry breaking mechanism? Widely studied ferromagnet/heavy metal bilayers (e.g., Co/Pt) also have the interfacial DMI having the same symmetry with the g-DMI of this work. Why does not the speculated mechanism work for Co/Pt? Is there any specific role of two sublattices of Co/Tb in the field-free switching?

Q3. Figure 1f shows that there is a local variation of concentration gradient in Co/Tb. For instance, both Co and Tb concentrations increase with the position Z for $11.5 \text{ nm} < Z < 14 \text{ nm}$, whereas the Co (Tb) concentration increases (decreases) for $14 \text{ nm} < Z < 17 \text{ nm}$. Then Co and Tb concentrations decrease for $17 \text{ nm} < Z < \sim 18 \text{ nm}$. This local variation of concentration ratio would result in a local variation of the magnitude and sign of both SOT and g-DMI. How does this local variation affect the field-free switching? Is it not a source of the symmetry breaking, instead of the speculated creation of a chiral magnetic texture?

A clear explanation about the symmetry breaking associated with satisfactory answers to all above questions must be given to deserve publication in Nature Communications. Without such clarity of the mechanism, I cannot recommend publication of this work. It is suited to a more specialized journal.

One minor comment:

In p4 of the main text, it was stated that "... may enable deterministic switching in the absence of an external field [33-35]. However, this type of field-free combined DMI-SOT switching has not been experimentally observed to date." This statement gives an impression that the current work experimentally proves the prediction in Refs. [33-35]. However, in p4 of the response letter, it was stated that "Note that, this behavior is qualitatively different from what Refs. [1.1-1.3] (Refs. [33-35] in the main text) predicted from micromagnetic simulations." These two statements contradict each other and thus must be corrected.

Reviewer #2:

Remarks to the Author:

The authors have addressed all my concerns in this revision. I now support this manuscript to be published in Nature Communications.

Reviewer #3:

Remarks to the Author:

I think that the authors have addressed the concerns and questions from reviewers, and that the manuscript is ready for publication.

Response Letter

COMMENTS TO AUTHOR:

Reviewer #1 (Remarks to the Author):

The revised manuscript includes additional switching experiments showing that the first current pulse (e.g., its polarity) has a definite role in the symmetry breaking and field-free switching. However, a concrete description about the symmetry breaking mechanism associated with the “first current pulse” is still ambiguous. The authors speculate that a creation of a chiral magnetic texture due to the simultaneous effect of SOT and g-DMI breaks the symmetry. To my view, a speculation about the underlying mechanism is not sufficient to be publishable in a high profile journal like Nature Communications.

Q1. Is it not possible to experimentally prove the creation of a chiral magnetic texture to break the symmetry?

Authors' Response:

Thank you for this suggestion. We performed new experiments, as described below, which provide additional evidence for the creation of a chiral magnetic texture (CMT) in response to training by a current pulse. Specifically, we hypothesize that if indeed

the presence of a CMT is responsible for the observed training effect, it should be possible to destroy the effect of this training using a sufficiently large magnetic field applied to the device. This, in turn, would allow one to subsequently *retrain* the device by applying a new training current pulse in the opposite direction. In addition, if this hypothesis is correct, there should be a range of smaller external magnetic fields which would not be sufficient to destroy the CMT, and hence the training effect. Our new experiments clearly confirmed this hypothesis, as described below:

- a) We chose a new device where no current pulses had been applied before. This device was first trained with a current pulse (0.1 ms, ~ 11 MA/cm²) **along the +x direction** (as defined in the coordinate axes shown in Fig. R1b).
- b) We then performed current-induced switching experiments in this device, using the 0 degree measurement configuration (Fig. R1b, top panel) and the 180 degree measurement configuration (Fig. R1b, bottom panel). Consistent with our previous results, the initial current pulse resulted in training of the device, giving rise to field-free deterministic and directional switching, with the 0 degree and 180 degree measurement configurations resulting in opposite polarities of the loops, as expected.
- c) We then applied an out-of-plane magnetic field H_1 to destroy any magnetic texture in the device, thus *resetting* the device to the initial pre-training state. This was followed by applying a current pulse (0.1 ms, ~ 11 MA/cm²) **along the -x direction** (as defined in the coordinate axes shown in Fig. R1b), to retrain the same device.
- d) Subsequent to this retraining attempt, we again performed current-induced switching measurements in the device using both the 0 degree measurement configuration (Fig. R1b, top panel) and the 180 degree measurement configuration (Fig. R1b, bottom panel).

The results of this *reset* and *retraining* experiment were as follows:

1. **Large H_1 field:** Fig. R1a shows the results for the case where the applied *reset* magnetic field H_1 is 8 T. Clearly, after going through an 8 T field and being retrained by a current along the -x direction, the device exhibits the opposite

switching directionality compared to the original training.

2. **Small H_1 field:** Next, we repeated the same sequence of experimental steps in another (also previously untrained) device, with the only difference being that the applied reset field H_1 was chosen to be smaller (0.2 T). Fig. R1c shows the results for this case, which clearly indicate that the *retraining* process does not work with this smaller field. In other words, the device keeps its original training and switching directionality after going through a 0.2 T field.

We interpret these results as a strong confirmation that a CMT is responsible for the observed training effect and directional switching in our devices. This CMT can be destroyed by a large field of 8 T, but not by a smaller field of 0.2 T.

Fig. R1 Device Retraining. **a**, Current-induced switching loops before and after re-initializing the device with an 8 T magnetic field and retraining with a current in the $-x$ direction. **b**, Schematic illustration of 0 degree and 180 degree measurement configurations. **c**, Current-induced switching loops before and after attempted re-initialization of the device with a 0.2 T magnetic field, which is not large enough to allow for retraining with a current in the $-x$ direction.

It is worth noting that in principle, even without applying an external 8 T field, a large enough current pulse in the opposite direction (compared to the initial training current) should be able to directly modify the CMT and retrain the device to exhibit an opposite switching directionality. In our experiments, however, we were not able to apply sufficiently large currents to do so without burning the device.

We also note that, while these experiments clearly indicate that a chiral magnetic texture is responsible for the field-free deterministic and directional switching observed in this work, we did not attempt to microscopically investigate or image the structure of these CMTs, which we believe goes beyond the scope of the present work. We hope that our findings will motivate subsequent work in the magnetism community to further elucidate the detailed microscopic nature of the CMTs and their effect on the magnetization switching characteristics.

Finally, it is worth noting that the concept of a directional anisotropy, defined by the history of applied fields or torques to the magnetization, has previously been observed in other material systems. Specifically, the “nonvolatile chirality printing” seen here is in some ways similar to the so-called “triad anisotropy”, previously studied in disordered magnetic alloys (spin glasses) exhibiting DMI induced by heavy impurities with large spin-orbit coupling [1.1-1.5]. In those cases, however, the emergence of a directional “triad anisotropy” was the result of the history of applied magnetic fields in conjunction with DMI, unlike our present case where the training occurs in response to SOT and DMI.

Indication of changes in the revised manuscript:

(1) Supplementary Note S9: We added text and Fig. S14 to describe the above-mentioned experiments.

Q2. Why is the Co/Tb special for this speculated symmetry breaking mechanism? Widely studied ferromagnet/heavy metal bilayers (e.g., Co/Pt) also have the interfacial DMI having the same symmetry with the g-DMI of this work. Why does not the speculated mechanism work for Co/Pt? Is there any specific role of two sublattices of Co/Tb in the field-free switching?

Authors' Response:

In principle, it should be possible to realize a similar field-free deterministic switching in conventional ferromagnet / heavy metal material systems, such as the Co/Pt system. However, there are reasons why it could be more difficult to do so compared to the

CoTb gradient material system presented here, which we summarize below.

First, it is worth noting that the simulation papers [1.6-1.8] which predicted DMI-induced field-free deterministic (but not directional) switching, all focused on heavy metal / ferromagnet bilayers. However, to achieve field-free switching in such systems, the switching dynamics required a strict balance between the interfacial DMI value and spin current density (in other words, the applied current density), which resulted in a very narrow operation window. This narrow operation window, which is highly material-dependent, could hinder the experimental realization of deterministic switching in heavy metal / ferromagnet bilayer systems.

Second, while it is in principle possible that interfacial DMI and SOT in an appropriately optimized Co/Pt-based material system could result in current-induced CMTs similar to our CoTb gradient samples (and thus also achieve directional field-free switching), this process could be more difficult due to the stronger dipolar interactions in ferromagnets [1.9-1.10]. Given that the interfacial DMI in systems like Co/Pt and the g-DMI in our ferrimagnetic samples are comparable, this difference in the magnetostatic dipole energy of ferromagnetic and ferrimagnetic systems could play an important role in whether chiral textures controlled by DMI can be stabilized during the training process.

Q3. Figure 1f shows that there is a local variation of concentration gradient in Co/Tb. For instance, both Co and Tb concentrations increase with the position Z for $11.5 \text{ nm} < Z < 14 \text{ nm}$, whereas the Co (Tb) concentration increases (decreases) for $14 \text{ nm} < Z < 17 \text{ nm}$. Then Co and Tb concentrations decrease for $17 \text{ nm} < Z < \sim 18 \text{ nm}$. This local variation of concentration ratio would result in a local variation of the magnitude and sign of both SOT and g-DMI. How does this local variation affect the field-free switching? Is it not a source of the symmetry breaking, instead of the speculated creation of a chiral magnetic texture?

Authors' Response:

Thank you for this observation. It is correct that the EDS intensities of both Co and Tb decrease as we approach the top/bottom of the CoTb gradient film. This is a result

of the inevitable diffusion and intermixing with the adjacent Al layers. However, we do not think that this intermixing can explain the experimental observations of this study, for the reasons discussed below.

First, we note that the type of symmetry breaking required to explain the field-free directional switching is an *in-plane* breaking of the inversion symmetry, which, as noted before, can be created by chiral magnetic textures. The intermixing of Co and Tb with adjacent layers only results in an overall reduction of the *out-of-plane* (vertical) composition gradient in the film. Thus, while it weakens the symmetry breaking along the growth (i.e., vertical) direction, it cannot by itself explain the emergence of field-free directional switching in the device. However, it should indeed be possible to further increase the magnitude of g-DMI and SOT, and thus the switching efficiency of our devices, by developing deposition and processing techniques that suppress this intermixing effect. We hope that this work will motivate further research towards the development of such optimized g-DMI material structures.

Second, we note that the observed field-free directional switching is also not a result of any unintentional in-plane symmetry breaking, as already demonstrated in Supplementary Note 9 through measurements on multiple devices with different orientations on the same chip.

Last but not least, we performed additional experiments to investigate the device-to-device uniformity not only within a single chip, but also across a wafer. We performed field-free switching experiments in devices randomly picked on a full 100 mm (4 inch) silicon wafer. The results are shown in Fig. R2, where parts 2, 4, 6 and 8 on the wafer (indicated in Fig. R2a) were selected for measurements. Fig. R2b plots the field-free current-induced switching loops of the devices on these different chips using the same measurement setup. Clearly, all measurements show the same switching polarity as well as similar critical switching currents. This device-to-device uniformity over a large area further reveals that very little in-plane concentration variations can exist in our samples. These results are also encouraging for the translation of this field-free SOT switching scheme to industrial manufacturing on

larger wafer sizes.

Fig. R2 Field free switching experiments on a full 100 mm wafer. a, Die layout and numbering of a 100 mm (4 inch) wafer. **b,** Current-induced SOT switching loops of devices in parts 2, 4, 6 and 8 on the wafer under zero in-plane field. All the devices on the wafer show the same switching behavior, indicating good device-to-device uniformity across the wafer.

Indication of changes in the revised manuscript:

- (1) The results of Fig. R2, along with the related discussion, were added in a new Supplementary Note S10.
- (2) In the main text, page 13, we added a reference to the new Supplementary Note 10, along with the sentence “This behavior was consistent for multiple devices measured across different locations on a 100 mm wafer.”

A clear explanation about the symmetry breaking associated with satisfactory answers to all above questions must be given to deserve publication in Nature Communications. Without such clarity of the mechanism, I cannot recommend publication of this work. It is suited to a more specialized journal.

One minor comment:

In p4 of the main text, it was stated that “... may enable deterministic switching in the absence of an external field [33-35]. However, this type of field-free combined

DMI-SOT switching has not been experimentally observed to date.” This statement gives an impression that the current work experimentally proves the prediction in Refs. [33-35]. However, in p4 of the response letter, it was stated that “Note that, this behavior is qualitatively different from what Refs. [1.1-1.3] (Refs. [33-35] in the main text) predicted from micromagnetic simulations.” These two statements contradict each other and thus must be corrected.

Authors’ Response:

Thank you for pointing this out. We have modified the corresponding sentence on page 4 to address this point.

Indication of changes in the revised manuscript:

Main text, page 4: “However, this type of field-free combined DMI-SOT switching has not been experimentally observed to date” was replaced by “However, field-free deterministic switching due to the combined action of DMI and SOT has not been reported to date.”

References

- [1.1]Fert, A. & Hippert, F. Anisotropy of spin-glasses from torque measurements. *Phys. Rev. Lett.* **49**, 1508 (1982).
- [1.2]Fert, A., Arvanitis, D. & Hippert, F. Triad anisotropy of spin glasses and torque experiments. *J. Appl. Phys.* **55**, 1640-1645 (1984).
- [1.3]Fert, A. & Levy, P. M. Role of anisotropic exchange interactions in determining the properties of spin-glasses. *Phys. Rev. Lett.* **44**, 1538 (1980).
- [1.4]Levy, P. M. & Fert, A. Anisotropy induced by nonmagnetic impurities in CuMn spin-glass alloys. *Phys. Rev. B* **23**, 4667 (1981).
- [1.5]Hippert, F., Alloul, H. & Fert, A. Anisotropy energy of CuMn spin glasses. *J. Appl. Phys.* **53**, 7702-7704 (1982).
- [1.6]Chen, B., Lourembam, J., Goolaup, S. & Lim, S. T. Field-free spin-orbit torque switching of a perpendicular ferromagnet with Dzyaloshinskii-Moriya interaction. *Appl. Phys. Lett.* **114**, 022401 (2019).

- [1.7] Wu, K., Su, D., Saha, R. & Wang, J. P. Deterministic field-free switching of a perpendicularly magnetized ferromagnetic layer via the joint effects of the Dzyaloshinskii–Moriya interaction and damping-and field-like spin–orbit torques: an appraisal. *J. Phys. D Appl. Phys.* **53**, 205002 (2020).
- [1.8] Dai, M. & Hu, J. M. Field-free spin–orbit torque perpendicular magnetization switching in ultrathin nanostructures. *NPJ Comput. Mater.* **6**, 1-10 (2020).
- [1.9] Legrand, W. et al. Room-temperature stabilization of antiferromagnetic skyrmions in synthetic antiferromagnets. *Nat. Mater.* **19**, 34-42 (2020).
- [1.10] Caretta, L. et al. Fast current-driven domain walls and small skyrmions in a compensated ferrimagnet. *Nat. Nanotechnol.* **13**, 1154-1160 (2018).

Reviewer #2 (Remarks to the Author):

The authors have addressed all my concerns in this revision. I now support this manuscript to be published in Nature Communications.

Authors' Response:

Thank you again for your time and for recommending this work for publication in Nature Communications.

Reviewer #3 (Remarks to the Author):

I think that the authors have addressed the concerns and questions from reviewers, and that the manuscript is ready for publication.

Authors' Response:

Thank you again for your time and for recommending this work for publication in Nature Communications.

Other modifications:

- 1) We found that the description of measurement angles in the previously revised version might cause misunderstanding for some readers. To avoid any confusion, we added electrical pad numbers for the device in the schematics shown in Figs. S11, S12, S13 and S14. Corresponding information was also added in Supplementary Note S9.
- 2) We add a new reference “[56] Cutugno, F. et al. Micromagnetic understanding of switching and self-oscillations in ferrimagnetic materials. *Appl. Phys. Lett.* **118**, 052403 (2021)” to our main text. Corresponding reference number modifications are also added in main text.

Reviewers' Comments:

Reviewer #1:

Remarks to the Author:

The authors' responses with additional experiments are satisfactory and my concern about the underlying mechanism of field-free switching is now resolved. I support publication of this work in Nature Communications.